# InfoGAN: Interpretable Representation Learning by Information Maximizing Generative Adversarial Nets

**Xi Chen**[†‡]**, Yan Duan**[†‡]**, Rein Houthooft**[†‡]**, John Schulman**[†‡]**, Ilya Sutskever**[‡]**, Pieter Abbeel**[†‡]

† UC Berkeley, Department of Electrical Engineering and Computer Sciences

‡ OpenAI

## Abstract

This paper describes InfoGAN, an information-theoretic extension to the Generative Adversarial Network that is able to learn disentangled representations in a completely unsupervised manner. InfoGAN is a generative adversarial network that also maximizes the mutual information between a small subset of the latent variables and the observation. We derive a lower bound of the mutual information objective that can be optimized efficiently. Specifically, InfoGAN successfully disentangles writing styles from digit shapes on the MNIST dataset, pose from lighting of 3D rendered images, and background digits from the central digit on the SVHN dataset. It also discovers visual concepts that include hair styles, presence/absence of eyeglasses, and emotions on the CelebA face dataset. Experiments show that InfoGAN learns interpretable representations that are competitive with representations learned by existing supervised methods. For an up-to-date version of this paper, please see https://arxiv.org/abs/1606.03657.

## 1 Introduction

Unsupervised learning can be described as the general problem of extracting value from unlabelled data which exists in vast quantities. A popular framework for unsupervised learning is that of representation learning [1, 2], whose goal is to use unlabelled data to learn a representation that exposes important semantic features as easily decodable factors. A method that can learn such representations is likely to exist [2], and to be useful for many downstream tasks which include classification, regression, visualization, and policy learning in reinforcement learning.

While unsupervised learning is ill-posed because the relevant downstream tasks are unknown at training time, a *disentangled representation*, one which explicitly represents the salient attributes of a data instance, should be helpful for the *relevant* but unknown tasks. For example, for a dataset of faces, a useful disentangled representation may allocate a separate set of dimensions for each of the following attributes: facial expression, eye color, hairstyle, presence or absence of eyeglasses, and the identity of the corresponding person. A disentangled representation can be useful for natural tasks that require knowledge of the salient attributes of the data, which include tasks like face recognition and object recognition. It is not the case for unnatural supervised tasks, where the goal could be, for example, to determine whether the number of red pixels in an image is even or odd. Thus, to be useful, an unsupervised learning algorithm must in effect correctly guess the likely set of downstream classification tasks without being directly exposed to them.

A significant fraction of unsupervised learning research is driven by generative modelling. It is motivated by the belief that the ability to synthesize, or "create" the observed data entails some form of understanding, and it is hoped that a good generative model will automatically learn a disentangled representation, even though it is easy to construct perfect generative models with arbitrarily bad representations. The most prominent generative models are the variational autoencoder (VAE) [3] and the generative adversarial network (GAN) [4].

In this paper, we present a simple modification to the generative adversarial network objective that encourages it to learn interpretable and meaningful representations. We do so by maximizing the mutual information between a fixed small subset of the GAN's noise variables and the observations, which turns out to be relatively straightforward. Despite its simplicity, we found our method to be surprisingly effective: it was able to discover highly semantic and meaningful hidden representations on a number of image datasets: digits (MNIST), faces (CelebA), and house numbers (SVHN). The quality of our unsupervised disentangled representation matches previous works that made use of supervised label information [5–9]. These results suggest that generative modelling augmented with a mutual information cost could be a fruitful approach for learning disentangled representations.

In the remainder of the paper, we begin with a review of the related work, noting the supervision that is required by previous methods that learn disentangled representations. Then we review GANs, which is the basis of InfoGAN. We describe how maximizing mutual information results in interpretable representations and derive a simple and efficient algorithm for doing so. Finally, in the experiments section, we first compare InfoGAN with prior approaches on relatively clean datasets and then show that InfoGAN can learn interpretable representations on complex datasets where no previous unsupervised approach is known to learn representations of comparable quality.

## 2   Related Work

There exists a large body of work on unsupervised representation learning. Early methods were based on stacked (often denoising) autoencoders or restricted Boltzmann machines [10–13].

Another intriguing line of work consists of the ladder network [14], which has achieved spectacular results on a semi-supervised variant of the MNIST dataset. More recently, a model based on the VAE has achieved even better semi-supervised results on MNIST [15]. GANs [4] have been used by Radford et al. [16] to learn an image representation that supports basic linear algebra on code space. Lake et al. [17] have been able to learn representations using probabilistic inference over Bayesian programs, which achieved convincing one-shot learning results on the OMNI dataset.

In addition, prior research attempted to learn disentangled representations using supervised data. One class of such methods trains a subset of the representation to match the supplied label using supervised learning: bilinear models [18] separate style and content; multi-view perceptron [19] separate face identity and view point; and Yang *et al.* [20] developed a recurrent variant that generates a sequence of latent factor transformations. Similarly, VAEs [5] and Adversarial Autoencoders [9] were shown to learn representations in which class label is separated from other variations.

Recently several weakly supervised methods were developed to remove the need of explicitly labeling variations. disBM [21] is a higher-order Boltzmann machine which learns a disentangled representation by "clamping" a part of the hidden units for a pair of data points that are known to match in all but one factors of variation. DC-IGN [7] extends this "clamping" idea to VAE and successfully learns graphics codes that can represent pose and light in 3D rendered images. This line of work yields impressive results, but they rely on a supervised grouping of the data that is generally not available. Whitney *et al.* [8] proposed to alleviate the grouping requirement by learning from consecutive frames of images and use temporal continuity as supervisory signal.

Unlike the cited prior works that strive to recover disentangled representations, InfoGAN requires no supervision of any kind. To the best of our knowledge, the only other unsupervised method that learns disentangled representations is hossRBM [13], a higher-order extension of the spike-and-slab restricted Boltzmann machine that can disentangle emotion from identity on the Toronto Face Dataset [22]. However, hossRBM can only disentangle discrete latent factors, and its computation cost grows exponentially in the number of factors. InfoGAN can disentangle both discrete and continuous latent factors, scale to complicated datasets, and typically requires no more training time than regular GANs.

## 3   Background: Generative Adversarial Networks

Goodfellow *et al.* [4] introduced the Generative Adversarial Networks (GAN), a framework for training deep generative models using a minimax game. The goal is to learn a generator distribution $P_G(x)$ that matches the real data distribution $P_{\text{data}}(x)$. Instead of trying to explicitly assign probability to every $x$ in the data distribution, GAN learns a generator network $G$ that generates samples from

the generator distribution $P_G$ by transforming a noise variable $z \sim P_{\text{noise}}(z)$ into a sample $G(z)$. This generator is trained by playing against an adversarial discriminator network $D$ that aims to distinguish between samples from the true data distribution $P_{\text{data}}$ and the generator's distribution $P_G$. So for a given generator, the optimal discriminator is $D(x) = P_{\text{data}}(x)/(P_{\text{data}}(x) + P_G(x))$. More formally, the minimax game is given by the following expression:

$$\min_G \max_D V(D, G) = \mathbb{E}_{x \sim P_{\text{data}}}[\log D(x)] + \mathbb{E}_{z \sim \text{noise}}[\log(1 - D(G(z)))] \tag{1}$$

## 4 Mutual Information for Inducing Latent Codes

The GAN formulation uses a simple factored continuous input noise vector $z$, while imposing no restrictions on the manner in which the generator may use this noise. As a result, it is possible that the noise will be used by the generator in a highly entangled way, causing the individual dimensions of $z$ to not correspond to semantic features of the data.

However, many domains naturally decompose into a set of semantically meaningful factors of variation. For instance, when generating images from the MNIST dataset, it would be ideal if the model automatically chose to allocate a discrete random variable to represent the numerical identity of the digit (0-9), and chose to have two additional continuous variables that represent the digit's angle and thickness of the digit's stroke. It would be useful if we could recover these concepts without any supervision, by simply specifying that an MNIST digit is generated by an 1-of-10 variable and two continuous variables.

In this paper, rather than using a single unstructured noise vector, we propose to decompose the input noise vector into two parts: (i) $z$, which is treated as source of incompressible noise; (ii) $c$, which we will call the latent code and will target the salient structured semantic features of the data distribution.

Mathematically, we denote the set of structured latent variables by $c_1, c_2, \ldots, c_L$. In its simplest form, we may assume a factored distribution, given by $P(c_1, c_2, \ldots, c_L) = \prod_{i=1}^{L} P(c_i)$. For ease of notation, we will use latent codes $c$ to denote the concatenation of all latent variables $c_i$.

We now propose a method for discovering these latent factors in an unsupervised way: we provide the generator network with both the incompressible noise $z$ and the latent code $c$, so the form of the generator becomes $G(z, c)$. However, in standard GAN, the generator is free to ignore the additional latent code $c$ by finding a solution satisfying $P_G(x|c) = P_G(x)$. To cope with the problem of trivial codes, we propose an information-theoretic regularization: there should be high mutual information between latent codes $c$ and generator distribution $G(z, c)$. Thus $I(c; G(z, c))$ should be high.

In information theory, mutual information between $X$ and $Y$, $I(X; Y)$, measures the "amount of information" learned from knowledge of random variable $Y$ about the other random variable $X$. The mutual information can be expressed as the difference of two entropy terms:

$$I(X; Y) = H(X) - H(X|Y) = H(Y) - H(Y|X) \tag{2}$$

This definition has an intuitive interpretation: $I(X; Y)$ is the reduction of uncertainty in $X$ when $Y$ is observed. If $X$ and $Y$ are independent, then $I(X; Y) = 0$, because knowing one variable reveals nothing about the other; by contrast, if $X$ and $Y$ are related by a deterministic, invertible function, then maximal mutual information is attained. This interpretation makes it easy to formulate a cost: given any $x \sim P_G(x)$, we want $P_G(c|x)$ to have a small entropy. In other words, the information in the latent code $c$ should not be lost in the generation process. Similar mutual information inspired objectives have been considered before in the context of clustering [23–25]. Therefore, we propose to solve the following information-regularized minimax game:

$$\min_G \max_D V_I(D, G) = V(D, G) - \lambda I(c; G(z, c)) \tag{3}$$

## 5 Variational Mutual Information Maximization

In practice, the mutual information term $I(c; G(z, c))$ is hard to maximize directly as it requires access to the posterior $P(c|x)$. Fortunately we can obtain a lower bound of it by defining an auxiliary

distribution $Q(c|x)$ to approximate $P(c|x)$:

$$
\begin{aligned}
I(c; G(z, c)) &= H(c) - H(c|G(z, c)) \\
&= \mathbb{E}_{x \sim G(z,c)}[\mathbb{E}_{c' \sim P(c|x)}[\log P(c'|x)]] + H(c) \\
&= \mathbb{E}_{x \sim G(z,c)}[\underbrace{D_{\mathrm{KL}}(P(\cdot|x) \parallel Q(\cdot|x))}_{\geq 0} + \mathbb{E}_{c' \sim P(c|x)}[\log Q(c'|x)]] + H(c) \quad (4) \\
&\geq \mathbb{E}_{x \sim G(z,c)}[\mathbb{E}_{c' \sim P(c|x)}[\log Q(c'|x)]] + H(c)
\end{aligned}
$$

This technique of lower bounding mutual information is known as Variational Information Maximization [26]. We note that the entropy of latent codes $H(c)$ can be optimized as well since it has a simple analytical form for common distributions. However, in this paper we opt for simplicity by fixing the latent code distribution and we will treat $H(c)$ as a constant. So far we have bypassed the problem of having to compute the posterior $P(c|x)$ explicitly via this lower bound but we still need to be able to sample from the posterior in the inner expectation. Next we state a simple lemma, with its proof deferred to Appendix 1, that removes the need to sample from the posterior.

**Lemma 5.1** *For random variables $X, Y$ and function $f(x, y)$ under suitable regularity conditions:* $\mathbb{E}_{x \sim X, y \sim Y|x}[f(x, y)] = \mathbb{E}_{x \sim X, y \sim Y|x, x' \sim X|y}[f(x', y)]$.

By using Lemma 5.1, we can define a variational lower bound, $L_I(G, Q)$, of the mutual information, $I(c; G(z, c))$:

$$
\begin{aligned}
L_I(G, Q) &= E_{c \sim P(c), x \sim G(z,c)}[\log Q(c|x)] + H(c) \\
&= E_{x \sim G(z,c)}[\mathbb{E}_{c' \sim P(c|x)}[\log Q(c'|x)]] + H(c) \quad (5) \\
&\leq I(c; G(z, c))
\end{aligned}
$$

We note that $L_I(G, Q)$ is easy to approximate with Monte Carlo simulation. In particular, $L_I$ can be maximized w.r.t. $Q$ directly and w.r.t. $G$ via the reparametrization trick. Hence $L_I(G, Q)$ can be added to GAN's objectives with no change to GAN's training procedure and we call the resulting algorithm Information Maximizing Generative Adversarial Networks (InfoGAN).

Eq (4) shows that the lower bound becomes tight as the auxiliary distribution $Q$ approaches the true posterior distribution: $\mathbb{E}_x[D_{\mathrm{KL}}(P(\cdot|x) \parallel Q(\cdot|x))] \to 0$. In addition, we know that when the variational lower bound attains its maximum $L_I(G, Q) = H(c)$ for discrete latent codes, the bound becomes tight and the maximal mutual information is achieved. In Appendix, we note how InfoGAN can be connected to the Wake-Sleep algorithm [27] to provide an alternative interpretation.

Hence, InfoGAN is defined as the following minimax game with a variational regularization of mutual information and a hyperparameter $\lambda$:

$$
\min_{G,Q} \max_D V_{\mathrm{InfoGAN}}(D, G, Q) = V(D, G) - \lambda L_I(G, Q) \quad (6)
$$

## 6 Implementation

In practice, we parametrize the auxiliary distribution $Q$ as a neural network. In most experiments, $Q$ and $D$ share all convolutional layers and there is one final fully connected layer to output parameters for the conditional distribution $Q(c|x)$, which means InfoGAN only adds a negligible computation cost to GAN. We have also observed that $L_I(G, Q)$ always converges faster than normal GAN objectives and hence InfoGAN essentially comes for *free* with GAN.

For categorical latent code $c_i$, we use the natural choice of softmax nonlinearity to represent $Q(c_i|x)$. For continuous latent code $c_j$, there are more options depending on what is the true posterior $P(c_j|x)$. In our experiments, we have found that simply treating $Q(c_j|x)$ as a factored Gaussian is sufficient.

Since GAN is known to be difficult to train, we design our experiments based on existing techniques introduced by DC-GAN [16], which are enough to stabilize InfoGAN training and we did not have to introduce new trick. Detailed experimental setup is described in Appendix. Even though InfoGAN introduces an extra hyperparameter $\lambda$, it's easy to tune and simply setting to 1 is sufficient for discrete latent codes. When the latent code contains continuous variables, a smaller $\lambda$ is typically used to ensure that $\lambda L_I(G, Q)$, which now involves differential entropy, is on the same scale as GAN objectives.

# 7 Experiments

The first goal of our experiments is to investigate if mutual information can be maximized efficiently. The second goal is to evaluate if InfoGAN can learn disentangled and interpretable representations by making use of the generator to vary only one latent factor at a time in order to assess if varying such factor results in only one type of semantic variation in generated images. DC-IGN [7] also uses this method to evaluate their learned representations on 3D image datasets, on which we also apply InfoGAN to establish direct comparison.

## 7.1 Mutual Information Maximization

To evaluate whether the mutual information between latent codes $c$ and generated images $G(z, c)$ can be maximized efficiently with proposed method, we train InfoGAN on MNIST dataset with a uniform categorical distribution on latent codes $c \sim \text{Cat}(K = 10, p = 0.1)$. In Fig 1, the lower bound $L_I(G, Q)$ is quickly maximized to $H(c) \approx 2.30$, which means the bound (4) is tight and maximal mutual information is achieved.

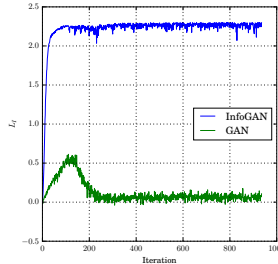

Figure 1: Lower bound $L_I$ over training iterations

As a baseline, we also train a regular GAN with an auxiliary distribution $Q$ when the generator is not explicitly encouraged to maximize the mutual information with the latent codes. Since we use expressive neural network to parametrize $Q$, we can assume that $Q$ reasonably approximates the true posterior $P(c|x)$ and hence there is little mutual information between latent codes and generated images in regular GAN. We note that with a different neural network architecture, there might be a higher mutual information between latent codes and generated images even though we have not observed such case in our experiments. This comparison is meant to demonstrate that in a regular GAN, there is no guarantee that the generator will make use of the latent codes.

## 7.2 Disentangled Representation

To disentangle digit shape from styles on MNIST, we choose to model the latent codes with one categorical code, $c_1 \sim \text{Cat}(K = 10, p = 0.1)$, which can model discontinuous variation in data, and two continuous codes that can capture variations that are continuous in nature: $c_2, c_3 \sim \text{Unif}(-1, 1)$.

In Figure 2, we show that the discrete code $c_1$ captures drastic change in shape. Changing categorical code $c_1$ switches between digits most of the time. In fact even if we just train InfoGAN without any label, $c_1$ can be used as a classifier that achieves 5% error rate in classifying MNIST digits by matching each category in $c_1$ to a digit type. In the second row of Figure 2a, we can observe a digit 7 is classified as a 9.

Continuous codes $c_2, c_3$ capture continuous variations in style: $c_2$ models rotation of digits and $c_3$ controls the width. What is remarkable is that in both cases, the generator does not simply stretch or rotate the digits but instead adjust other details like thickness or stroke style to make sure the resulting images are natural looking. As a test to check whether the latent representation learned by InfoGAN is generalizable, we manipulated the latent codes in an *exaggerated* way: instead of plotting latent codes from $-1$ to $1$, we plot it from $-2$ to $2$ covering a wide region that the network was never trained on and we still get meaningful generalization.

Next we evaluate InfoGAN on two datasets of 3D images: faces [28] and chairs [29], on which DC-IGN was shown to learn highly interpretable graphics codes.

On the faces dataset, DC-IGN learns to represent latent factors as azimuth (pose), elevation, and lighting as continuous latent variables by using supervision. Using the same dataset, we demonstrate that InfoGAN learns a disentangled representation that recover azimuth (pose), elevation, and lighting on the same dataset. In this experiment, we choose to model the latent codes with five continuous codes, $c_i \sim \text{Unif}(-1, 1)$ with $1 \leq i \leq 5$.

Since DC-IGN requires supervision, it was previously not possible to learn a latent code for a variation that's unlabeled and hence salient latent factors of variation cannot be discovered automatically from data. By contrast, InfoGAN is able to discover such variation on its own: for instance, in Figure 3d a

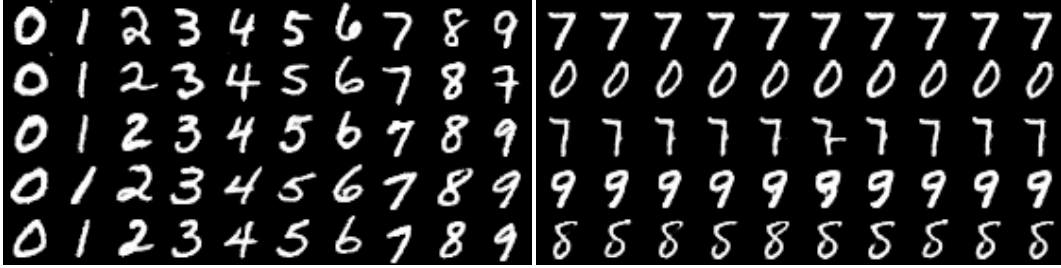

(a) Varying $c_1$ on InfoGAN (Digit type)          (b) Varying $c_1$ on regular GAN (No clear meaning)

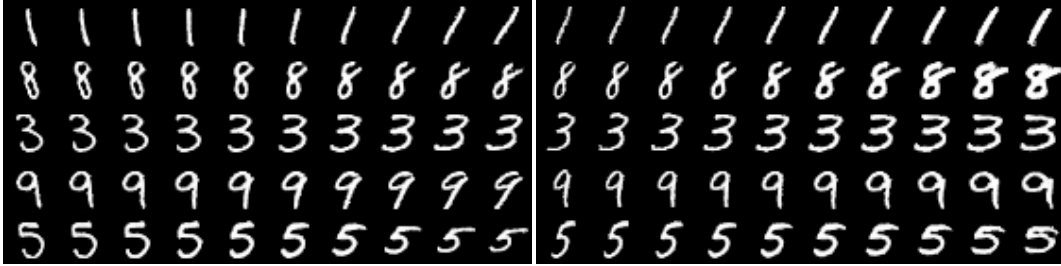

(c) Varying $c_2$ from $-2$ to 2 on InfoGAN (Rotation)          (d) Varying $c_3$ from $-2$ to 2 on InfoGAN (Width)

Figure 2: **Manipulating latent codes on MNIST:** *In all figures of latent code manipulation, we will use the convention that in each one latent code varies from left to right while the other latent codes and noise are fixed. The different rows correspond to different random samples of fixed latent codes and noise. For instance, in (a), one column contains five samples from the same category in $c_1$, and a row shows the generated images for 10 possible categories in $c_1$ with other noise fixed.* In (a), each category in $c_1$ largely corresponds to one digit type; in (b), varying $c_1$ on a GAN trained without information regularization results in non-interpretable variations; in (c), a small value of $c_2$ denotes left leaning digit whereas a high value corresponds to right leaning digit; in (d), $c_3$ smoothly controls the width. We reorder (a) for visualization purpose, as the categorical code is inherently unordered.

latent code that smoothly changes a face from wide to narrow is learned even though this variation was neither explicitly generated or labeled in prior work.

On the chairs dataset, DC-IGN can learn a continuous code that represents rotation. InfoGAN again is able to learn the same concept as a continuous code (Figure 4a) and we show in addition that InfoGAN is also able to continuously interpolate between similar chair types of different widths using a single continuous code (Figure 4b). In this experiment, we choose to model the latent factors with four categorical codes, $c_{1,2,3,4} \sim \text{Cat}(K = 20, p = 0.05)$ and one continuous code $c_5 \sim \text{Unif}(-1, 1)$.

Next we evaluate InfoGAN on the Street View House Number (SVHN) dataset, which is significantly more challenging to learn an interpretable representation because it is noisy, containing images of variable-resolution and distracting digits, and it does not have multiple variations of the same object. In this experiment, we make use of four $10-$dimensional categorical variables and two uniform continuous variables as latent codes. We show two of the learned latent factors in Figure 5.

Finally we show in Figure 6 that InfoGAN is able to learn many visual concepts on another challenging dataset: CelebA [30], which includes $200,000$ celebrity images with large pose variations and background clutter. In this dataset, we model the latent variation as 10 uniform categorical variables, each of dimension 10. Surprisingly, even in this complicated dataset, InfoGAN can recover azimuth as in 3D images even though in this dataset no single face appears in multiple pose positions. Moreover InfoGAN can disentangle other highly semantic variations like presence or absence of glasses, hairstyles and emotion, demonstrating a level of visual understanding is acquired.

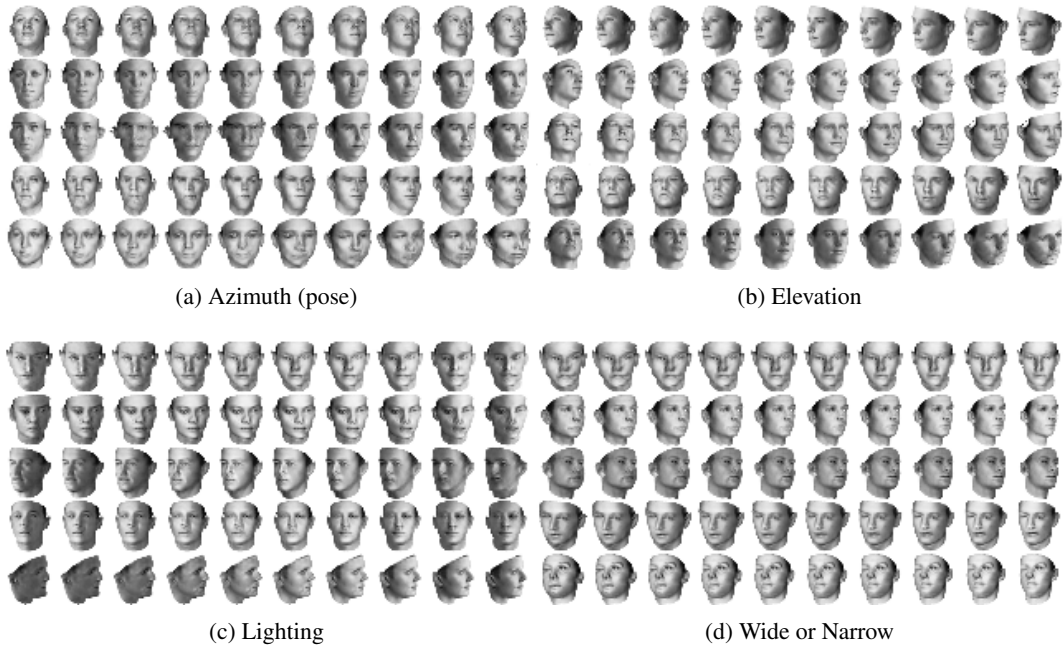

|  |  |
|---|---|
| (a) Azimuth (pose) | (b) Elevation |
| (c) Lighting | (d) Wide or Narrow |

Figure 3: **Manipulating latent codes on 3D Faces:** We show the effect of the learned continuous latent factors on the outputs as their values vary from $-1$ to $1$. In (a), we show that a continuous latent code consistently captures the azimuth of the face across different shapes; in (b), the continuous code captures elevation; in (c), the continuous code captures the orientation of lighting; and in (d), the continuous code learns to interpolate between wide and narrow faces while preserving other visual features. For each factor, we present the representation that most resembles prior results [7] out of 5 random runs to provide direct comparison.

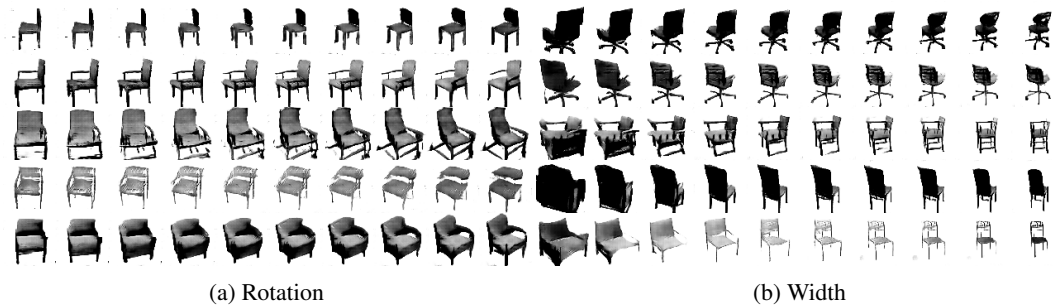

|  |  |
|---|---|
| (a) Rotation | (b) Width |

Figure 4: **Manipulating latent codes on 3D Chairs:** In (a), the continuous code captures the pose of the chair while preserving its shape, although the learned pose mapping varies across different types; in (b), the continuous code can alternatively learn to capture the widths of different chair types, and smoothly interpolate between them. For each factor, we present the representation that most resembles prior results [7] out of 5 random runs to provide direct comparison.

# 8 Conclusion

This paper introduces a representation learning algorithm called Information Maximizing Generative Adversarial Networks (InfoGAN). In contrast to previous approaches, which require supervision, InfoGAN is completely unsupervised and learns interpretable and disentangled representations on challenging datasets. In addition, InfoGAN adds only negligible computation cost on top of GAN and is easy to train. The core idea of using mutual information to induce representation can be applied to other methods like VAE [3], which is a promising area of future work. Other possible extensions to this work include: learning hierarchical latent representations, improving semi-supervised learning with better codes [31], and using InfoGAN as a high-dimensional data discovery tool.

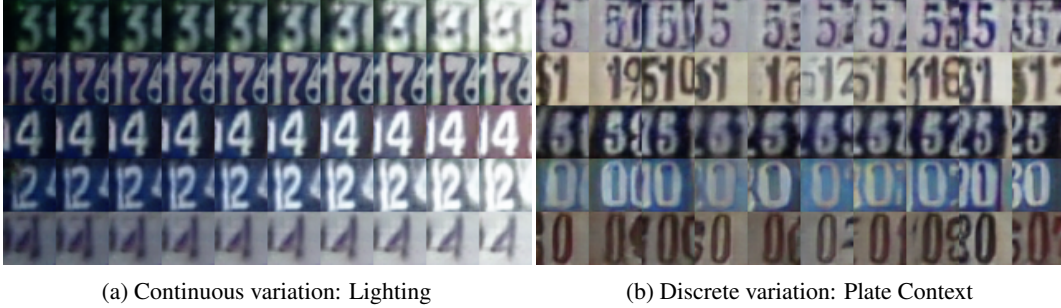

(a) Continuous variation: Lighting    (b) Discrete variation: Plate Context

Figure 5: **Manipulating latent codes on SVHN**: In (a), we show that one of the continuous codes captures variation in lighting even though in the dataset each digit is only present with one lighting condition; In (b), one of the categorical codes is shown to control the context of central digit: for example in the 2nd column, a digit 9 is (partially) present on the right whereas in 3rd column, a digit 0 is present, which indicates that InfoGAN has learned to separate central digit from its context.

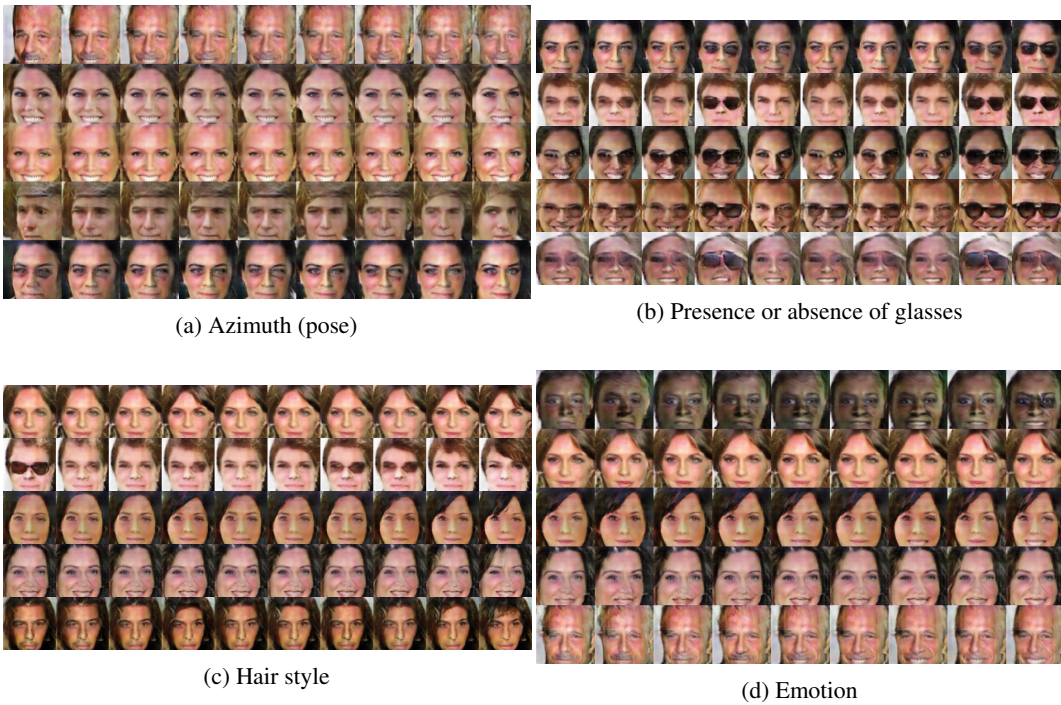

(a) Azimuth (pose)    (b) Presence or absence of glasses

(c) Hair style    (d) Emotion

Figure 6: **Manipulating latent codes on CelebA:** (a) shows that a categorical code can capture the azimuth of face by discretizing this variation of continuous nature; in (b) a subset of the categorical code is devoted to signal the presence of glasses; (c) shows variation in hair style, roughly ordered from less hair to more hair; (d) shows change in emotion, roughly ordered from stern to happy.

## Acknowledgements

We thank the anonymous reviewers. This research was funded in part by ONR through a PECASE award. Xi Chen was also supported by a Berkeley AI Research lab Fellowship. Yan Duan was also supported by a Berkeley AI Research lab Fellowship and a Huawei Fellowship. Rein Houthooft was supported by a Ph.D. Fellowship of the Research Foundation - Flanders (FWO).

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
