[Supplementary Material]

# Appendix of InfoGAN: Interpretable Representation Learning by Information Maximizing Generative Adversarial Nets

**Xi Chen**[†‡], **Yan Duan**[†‡], **Rein Houthooft**[†‡], **John Schulman**[†‡], **Ilya Sutskever**[‡], **Pieter Abbeel**[†‡]
† UC Berkeley, Department of Electrical Engineering and Computer Sciences
‡ OpenAI

## A Proof of Lemma 5.1

**Lemma A.1** *For random variables $X, Y$ and function $f(x, y)$ under suitable regularity conditions:* $\mathbb{E}_{x \sim X, y \sim Y|x}[f(x, y)] = \mathbb{E}_{x \sim X, y \sim Y|x, x' \sim X|y}[f(x', y)]$.

**Proof**

$$
\begin{aligned}
\mathbb{E}_{x \sim X, y \sim Y|x}[f(x, y)] &= \int_x P(x) \int_y P(y|x) f(x, y) dy dx \\
&= \int_x \int_y P(x, y) f(x, y) dy dx \\
&= \int_x \int_y P(x, y) f(x, y) \int_{x'} P(x'|y) dx' dy dx \\
&= \int_x P(x) \int_y P(y|x) \int_{x'} P(x'|y) f(x', y) dx' dy dx \\
&= \mathbb{E}_{x \sim X, y \sim Y|x, x' \sim X|y}[f(x', y)]
\end{aligned}
\tag{1}
$$

## B Interpretation as "Sleep-Sleep" Algorithm

We note that InfoGAN can be viewed as a Helmholtz machine [1]: $P_G(x|c)$ is the generative distribution and $Q(c|x)$ is the recognition distribution. Wake-Sleep algorithm [2] was proposed to train Helmholtz machines by performing "wake" phase and "sleep" phase updates.

The "wake" phase update proceeds by optimizing the variational lower bound of $\log P_G(x)$ w.r.t. generator:

$$
\max_G \mathbb{E}_{x \sim \text{Data}, c \sim Q(c|x)}[\log P_G(x|c)]
\tag{2}
$$

The "sleep" phase updates the auxiliary distribution $Q$ by "dreaming" up samples from current generator distribution rather than drawing from real data distribution:

$$
\max_Q \mathbb{E}_{c \sim P(c), x \sim P_G(x|c)}[\log Q(c|x)]
\tag{3}
$$

Hence we can see that when we optimize the surrogate loss $L_I$ w.r.t. $Q$, the update step is exactly the "sleep" phase update in Wake-Sleep algorithm. InfoGAN differs from Wake-Sleep when we optimize $L_I$ w.r.t. $G$, encouraging the generator network $G$ to make use of latent codes $c$ for the whole prior distribution on latent codes $P(c)$. Since InfoGAN also updates generator in "sleep" phase, our method can be interpreted as "Sleep-Sleep" algorithm. This interpretation highlights InfoGAN's difference from previous generative modeling techniques: the generator is explicitly encouraged to convey information in latent codes and suggests that the same principle can be applied to other generative models.

## C  Experiment Setup

For all experiments, we use Adam [3] for online optimization and apply batch normalization [4] after most layers, the details of which are specified for each experiment. We use an up-convolutional architecture for the generator networks [5]. We use leaky rectified linear units (lRELU) [6] with leaky rate 0.1 as the nonlinearity applied to hidden layers of the discrminator networks, and normal rectified linear units (RELU) for the generator networks. Unless noted otherwise, learning rate is 2e-4 for $D$ and 1e-3 for $G$; $\lambda$ is set to 1.

For discrete latent codes, we apply a softmax nonlinearity over the corresponding units in the recognition network output. For continuous latent codes, we parameterize the approximate posterior through a diagonal Gaussian distribution, and the recognition network outputs its mean and standard deviation, where the standard deviation is parameterized through an exponential transformation of the network output to ensure positivity.

The details for each set of experiments are presented below.

### C.1  MNIST

The network architectures are shown in Table 1. The discriminator $D$ and the recognition network $Q$ shares most of the network. For this task, we use 1 ten-dimensional categorical code, 2 continuous latent codes and 62 noise variables, resulting in a concatenated dimension of 74.

Table 1: The discriminator and generator CNNs used for MNIST dataset.

| discriminator $D$ / recognition network $Q$ | generator $G$ |
|---|---|
| Input $28 \times 28$ Gray image | Input $\in \mathbb{R}^{74}$ |
| $4 \times 4$ conv. 64 lRELU. stride 2 | FC. 1024 RELU. batchnorm |
| $4 \times 4$ conv. 128 lRELU. stride 2. batchnorm | FC. $7 \times 7 \times 128$ RELU. batchnorm |
| FC. 1024 lRELU. batchnorm | $4 \times 4$ upconv. 64 RELU. stride 2. batchnorm |
| FC. output layer for $D$, FC.128-batchnorm-lRELU-FC.output for $Q$ | $4 \times 4$ upconv. 1 channel |

### C.2  SVHN

The network architectures are shown in Table 2. The discriminator $D$ and the recognition network $Q$ shares most of the network. For this task, we use 4 ten-dimensional categorical code, 4 continuous latent codes and 124 noise variables, resulting in a concatenated dimension of 168.

Table 2: The discriminator and generator CNNs used for SVHN dataset.

| discriminator $D$ / recognition network $Q$ | generator $G$ |
|---|---|
| Input $32 \times 32$ Color image | Input $\in \mathbb{R}^{168}$ |
| $4 \times 4$ conv. 64 lRELU. stride 2 | FC. $2 \times 2 \times 448$ RELU. batchnorm |
| $4 \times 4$ conv. 128 lRELU. stride 2. batchnorm | $4 \times 4$ upconv. 256 RELU. stride 2. batchnorm |
| $4 \times 4$ conv. 256 lRELU. stride 2. batchnorm | $4 \times 4$ upconv. 128 RELU. stride 2. |
| FC. output layer for $D$, FC.128-batchnorm-lRELU-FC.output for $Q$ | $4 \times 4$ upconv. 64 RELU. stride 2. |
| | $4 \times 4$ upconv. 3 Tanh. stride 2. |

### C.3  CelebA

The network architectures are shown in Table 3. The discriminator $D$ and the recognition network $Q$ shares most of the network. For this task, we use 10 ten-dimensional categorical code and 128 noise variables, resulting in a concatenated dimension of 228.

Table 3: The discriminator and generator CNNs used for SVHN dataset.

| discriminator $D$ / recognition network $Q$ | generator $G$ |
|---|---|
| Input $32 \times 32$ Color image | Input $\in \mathbb{R}^{228}$ |
| $4 \times 4$ conv. 64 lRELU. stride 2 | FC. $2 \times 2 \times 448$ RELU. batchnorm |
| $4 \times 4$ conv. 128 lRELU. stride 2. batchnorm | $4 \times 4$ upconv. 256 RELU. stride 2. batchnorm |
| $4 \times 4$ conv. 256 lRELU. stride 2. batchnorm | $4 \times 4$ upconv. 128 RELU. stride 2. |
| FC. output layer for $D$, FC.128-batchnorm-lRELU-FC.output for $Q$ | $4 \times 4$ upconv. 64 RELU. stride 2. |
| | $4 \times 4$ upconv. 3 Tanh. stride 2. |

## C.4 Faces

The network architectures are shown in Table 4. The discriminator $D$ and the recognition network $Q$ shares the same network, and only have separate output units at the last layer. For this task, we use 5 continuous latent codes and 128 noise variables, so the input to the generator has dimension 133.

Table 4: The discriminator and generator CNNs used for Faces dataset.

| discriminator $D$ / recognition network $Q$ | generator $G$ |
|---|---|
| Input $32 \times 32$ Gray image | Input $\in \mathbb{R}^{133}$ |
| $4 \times 4$ conv. 64 lRELU. stride 2 | FC. 1024 RELU. batchnorm |
| $4 \times 4$ conv. 128 lRELU. stride 2. batchnorm | FC. $8 \times 8 \times 128$ RELU. batchnorm |
| FC. 1024 lRELU. batchnorm | $4 \times 4$ upconv. 64 RELU. stride 2. batchnorm |
| FC. output layer | $4 \times 4$ upconv. 1 sigmoid. |

We used separate configurations for each learned variation, shown in Table 5.

Table 5: The hyperparameters for Faces dataset.

| | Learning rate for $D$ / $Q$ | Learning rate for $G$ | $\lambda$ |
|---|---|---|---|
| Azimuth (pose) | 2e-4 | 5e-4 | 0.2 |
| Elevation | 4e-4 | 3e-4 | 0.1 |
| Lighting | 8e-4 | 3e-4 | 0.1 |
| Wide or Narrow | learned using the same network as the lighting variation | | |

## C.5 Chairs

The network architectures are shown in Table 6. The discriminator $D$ and the recognition network $Q$ shares the same network, and only have separate output units at the last layer. For this task, we use 1 continuous latent code, 3 discrete latent codes (each with dimension 20), and 128 noise variables, so the input to the generator has dimension 189.

We used separate configurations for each learned variation, shown in Table 7. For this task, we found it necessary to use different regularization coefficients for the continuous and discrete latent codes.

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

Table 6: The discriminator and generator CNNs used for Chairs dataset.

| discriminator $D$ / recognition network $Q$ | generator $G$ |
| --- | --- |
| Input $64 \times 64$ Gray image | Input $\in \mathbb{R}^{189}$ |
| $4 \times 4$ conv. 64 lRELU. stride 2 | FC. 1024 RELU. batchnorm |
| $4 \times 4$ conv. 128 lRELU. stride 2. batchnorm | FC. $8 \times 8 \times 256$ RELU. batchnorm |
| $4 \times 4$ conv. 256 lRELU. stride 2. batchnorm | $4 \times 4$ upconv. 256 RELU. batchnorm |
| $4 \times 4$ conv. 256 lRELU. batchnorm | $4 \times 4$ upconv. 256 RELU. batchnorm |
| $4 \times 4$ conv. 256 lRELU. batchnorm | $4 \times 4$ upconv. 128 RELU. stride 2. batchnorm |
| FC. 1024 lRELU. batchnorm | $4 \times 4$ upconv. 64 RELU. stride 2. batchnorm |
| FC. output layer | $4 \times 4$ upconv. 1 sigmoid. |

Table 7: The hyperparameters for Chairs dataset.

| | Learning rate for $D$ / $Q$ | Learning rate for $G$ | $\lambda_{\mathrm{cont}}$ | $\lambda_{\mathrm{disc}}$ |
| --- | --- | --- | --- | --- |
| Rotation | 2e-4 | 1e-3 | 10.0 | 1.0 |
| Width | 2e-4 | 1e-3 | 0.05 | 2.0 |

[5] A. Dosovitskiy, J. Tobias Springenberg, and T. Brox, "Learning to generate chairs with convolutional neural networks," in *Proceedings of the IEEE Conference on Computer Vision and Pattern Recognition*, 2015, pp. 1538–1546.

[6] A. L. Maas, A. Y. Hannun, and A. Y. Ng, "Rectifier nonlinearities improve neural network acoustic models," in *Proc. ICML*, vol. 30, 2013, p. 1.