[Reviews · NeurIPS 2016]

Reviewer 1

Summary

The paper presents an extension of the standard GAN framework, that allows to learn a set of disentangled interpretable codes in an unsupervised manner. The approach is motivated from the information-theoretic point of view and is based on minimizing the mutual information between the latent codes and the generated images. The experiments show the results on several types of image data: handwritten digits, house numbers, faces, chairs.

Qualitative Assessment

It is a good paper that should definitely be accepted. The presented approach has a clear theoretical motivation and is supported by a thorough and convincing experimental evaluation. It is important that the approach does not use any domain-specific knowledge and effectively comes at zero additional computational cost. This makes it easily applicable to a wide range of generative tasks. I have several questions/comments: 1) It seems to me that the proposed approach in the end amounts to training a GAN with an additional network (or an additional branch on top of the discriminator) trained to predict part of the latent code from the generated image. Is it correct? If not, what is the difference and how would such simple approach perform? If yes, please clearly state this fact in the paper. This interpretation is very intuitive and natural. And it is misleading to make it look more complicated than it is. Please address this question in the rebuttal. 2) All results are purely qualitative, except the remark on the 5% classification performance on MNIST. Some quantitative results, showing how the learned representation is useful for discriminative tasks, would strengthen the paper. 3) The paper only shows the effect of some selected latent factors on the generated image. It would be interesting to see the effect of all factors (in the supplementary, of course). Are they all meaningful? It would also be interesting to see the effect of "incompressible noise" z. 4) Why is "incompressible" noise z at all necessary? How do you select the dimensionality of z? 5) GANs are known to often fit only a subset of the data. Does InfoGAN have any advantage in this respect, or can it happen that it fits only a subset of the factors of variation? 6) For completeness it would be nice to have V(D,G) formulated in terms of x,c,z: it is not 100% clear from equation (1). 7) It would of course be interesting to see what kind of factors can the proposed method extract from more diverse data, for example the bedroom dataset used in the DCGAN paper or natural image datasets such as CIFAR-10 or ImageNet.

Confidence in this Review

2-Confident (read it all; understood it all reasonably well)


Reviewer 2

Summary

This paper presents an extension of GAN to learn interpretable representations (codes to generate images). The main observation is that the original GAN tends to ignore the additional condition (code) to generate images P(x|z) = P(x|z,c) where c is the code. To address this issue, this paper proposes to maximize the mutual information between the code c and the generated images G(z,c), and derive a variational lower bound for learning with a proposal posterior distribution Q(c|x). The final learning objective is a combination of minmax game loss of GAN and log likelihood of the proposal code distribution. In network implementation, this new code likelihood term is simply added at the end of discriminator with additional linear layers. Experiments are carried out on MNIST, renderings of faces and chairs, SVHN and CelebA with codes being categorical or uniform.

Qualitative Assessment

1. ----------------Technical quality: -The descriptions from Line 64 to 77 seem a bit confusing. What I understand is [22] and [23] are weakly-supervised methods while [24] and [7] are actually fully-supervised methods. -The comment in Line 79 seems to be inaccurate. Another notable unsupervised method for disentangling is the work by Cheung et al.: Discovering Hidden Factors of Variation in Deep Networks, ICLR 2015. -In Line 165, this paper claimed to use factorized Gaussian as code distribution, but it turned out that it used uniform distributions through all the experiments. This should be fixed. -One question about the formulation is about z, the free random vector. I wonder whether z could be considered as the same role as c so that the model only needs to model the mutual information between z and x. If not, what role is z playing in the G(z,c) model? -The proposed algorithm seems to achieve nice results on MNIST: different numbers are well separated in the categorical code. Interesting, the paper actually provides the knowledge about total number of categories for the code distribution (K). This information could be privileged in real-world. I wonder what results can be obtained if the K value is altered. -This paper presents many qualitative results on five datasets but little quantitative results (only Figure 1 and classification error on MNIST in Line 204). Quantitative evaluation of generated samples has been challenging, but considering the main argument of this paper is to learn disentangled representations for downstream applications, some classification experiments on faces, chairs and house numbers using learned features will be really valuable. 2. ----------------Novelty: +The information theory based derivation of learning disentangled codes is somewhat novel, although it bears similar theoretical background with wake-sleep algorithm and variational auto-encoders (VAEs) regarding joint learning of inference and generation networks. In VAEs, the sampling chain is image -> code -> image while the sampling chain used in this paper is code -> image -> code. Having code in the middle restricts the form of code distribution for enabling end-to-end training. However, in this paper having the code at the two ends of the network grants a flexible use of different code distributions such as categorical or uniform distributions. I believe this is the main advantage of the proposed algorithm. 3. ----------------Impact: +Learning interpretable and disentangled representations with generative models is an important research issue. The maximization of mutual information between code and image provides new insights and improves a popular generative model (GAN). I believe this has certain impact in academics. 4.-----------------Presentation:

Confidence in this Review

2-Confident (read it all; understood it all reasonably well)


Reviewer 3

Summary

The paper presents InfoGAN that adds information maximization term, which maximizes the information between the latent codes and the generated samples guided by the latent codes, to GAN objective function. By doing so, the latent codes can disentangle latent factors of variation in the training data without explicit supervision on those factors. To train with the mutual information maximization objective, the paper proposes the variational method that lower bounds the mutual information and the objective function becomes tractable. In experiments, the paper demonstrates the effectiveness of the model in disentangling interpretable visual factors using a few latent codes, such as azimuth, lighting, elevation in 3D synthesized face images.

Qualitative Assessment

The proposed regularization of InfoGAN is novel that assigns latent codes to contribute certain aspects of factors of variation in image generation. The formulation is technically sound and experimental results demonstrate the effectiveness of proposed algorithm well. I don't have much concern on this paper than some clarification on the training of InfoGAN. For the minimax optimization of G, Q, and D, in what order the parameters (or networks) are updated? Do you alternately update between G and Q, D? It seems so in this case because of shared parameters between Q and D, but what if the parameters of Q and D are not shared? Can G and Q updated at the same time?

Confidence in this Review

3-Expert (read the paper in detail, know the area, quite certain of my opinion)


Reviewer 4

Summary

The authors propose a new regularization of the GAN training objective. They maximize the mutual information I(c, G(c,z)), where z is the noise variable and c is a subset of the noise variables. The authors derive a lower bound on I(c,G(c,z) which can be optimized at low additional computational cost. The model is evaluated on 5 different datasets. The experiments are mainly qualitative but shows that the proposed regularization result in interesting latent codes. The effect is especially prominent in the MNIST experiments.

Qualitative Assessment

I find the paper pleasant to read. The idea of maximizing the mutual information is novel for GANs. My main concern is that the experiments are qualitative. The MNIST experiments shows some really nice learned features, but to me the results on SVHN and CelebA are less clear. In line 204 you mention that you get 5% error on MNIST by matching the latent code with the correct class. Maybe you could evaluate your method for classification on other datasets as well? In section 6 you write that you let D and Q share parameters. Does the method still work if you use a separate paramters for Q and D? In line 49-52 you mention that you compare with prior approaches? I cannot find that comparison? Minor suggestions ------------------ 1) equation (1): i think z~noise should be P_{noise} 2) Could you elborate a little more on the proof for lemma 5.1?

Confidence in this Review

2-Confident (read it all; understood it all reasonably well)


Reviewer 5

Summary

This paper extends GANs with an information theoretic criterion so as to produce disentangled representations. In particular, along with the usual GAN objective, infoGAN also maximizes mutual information between a small subset of latent variables and the observation. Latent z has an explicit structure -- incompressible noise and latent variables representing some semantic features of the data distribution. However, in pure GAN the learning algorithm will likely ignore this decomposition. There should be high mutual information between latent code and generator distribution from GAN. Alternatively, latent code should not be lost in the generative process. This way, the original GAN objective has one more term -- variational bound on the mutual information. This technique of lower bounding the mutual information was first proposed by Barber et al. and this paper successfully makes use of it. Once this term is added to the objective, training is similar to original GANs. InfoGAN shows impressive results on a wide variety of stimuli -- MNIST, 3D face, chairs, SVHN, CelebA (faces). The qualitative results highlight the fact that the model is able to disentangle multiple factors of intrinsic variations without supervision. I really like this paper and it should be definitely accepted. To the best of my knowledge, it makes meaningful scientific contributions towards building deep interpretable and disentangled representations.

Qualitative Assessment

- Overall this paper is very well written. However, to improve the clarity of the paper, authors should include a schematic illustrating their approach. Due to many pathways in the model, a pictorial illustration would be very helpful for many readers. - Are the representations *always* interpretable? Or do some of the runs produce entangled representations, especially if the latent code is not designed properly to reflect hidden semantic structure in the data distribution. If so, then the claim that the model learns interpretable representations via purely unsupervised learning might be a bit too strong. - Authors should discuss and highlight limitations. For instance, even in the results already presented (e.g. Fig 3(a)), the azimuth range seems biased towards a certain direction and does not cover the whole range. Or zooming into the same figure shows that face identity is sometimes changed even if only the azimuth is changed. - The qualitative results are very compelling. I would also like to see some quantitative results, especially on the chair and 3D face dataset. These datasets in particular seem to contain true pose/lighting values and it would be helpful to see at least a scatter plot showing inferred vs. predicted values. This seems important before the paper can be accepted.

Confidence in this Review

3-Expert (read the paper in detail, know the area, quite certain of my opinion)


Reviewer 6

Summary

This paper proposes a simple extension to generative adversarial networks motivated from information theory. The authors propose to condition the generator on a latent code which is learned to maximize the mutual information between the latent code and the generator. As a result, the model disentangles the factors of variation and learns interpretable latent representations in an unsupervised fashion.

Qualitative Assessment

This paper is one of the first deep learning paper that accomplishes to learn interpretable latent variables for the generative models in a completely unsupervised fashion. The extension that is provided by the authors is very simple and intuitive. The results are very impressive. The authors were able to show convincing results on the interpretability of the latent space. Is there any failure case of the model, e.g. any particular task or particular dataset where the model failed to perform well-enough? The paper is very well-written It would also be very interesting to report on how useful this model can be for pre-training of supervised models or the results for the semi-supervised learning setting which is left as a future work.

Confidence in this Review

2-Confident (read it all; understood it all reasonably well)